# The Asymmetric Laplace Gaussian (ALG) Distribution as the Descriptive Model for the Internal Proactive Inhibition in the Standard Stop Signal Task

**DOI:** 10.3390/brainsci12060730

**Published:** 2022-06-01

**Authors:** Mohsen Soltanifar, Michael Escobar, Annie Dupuis, Andre Chevrier, Russell Schachar

**Affiliations:** 1Biostatistics Division, Dalla Lana School of Public Health, University of Toronto, 620, 155 College Street, Toronto, ON M5T 3M7, Canada; m.escobar@utoronto.ca (M.E.); annie.dupuis@mdstats.ca (A.D.); 2The Hospital for Sick Children, Psychiatry Research, 4274, 4th Floor, Black Wing, 555 University Avenue, Toronto, ON M5G 1X8, Canada; andrechevrier135@gmail.com (A.C.); russell.schachar@sickkids.ca (R.S.); 3Continuing Studies Division, Population Data BC, University of Victoria, B364-3800 Finnerty Road, Victoria, BC V8P 5C2, Canada; 4Real World Analytics, Cytel Canada Health Inc., Vancouver, BC V5Z 1J5, Canada

**Keywords:** proactive inhibition, reaction times, Ex-Gaussian, Asymmetric Laplace Gaussian, Bayesian Parametric Approach, hazard function

## Abstract

Measurements of response inhibition components of reactive inhibition and proactive inhibition within the stop-signal paradigm have been of particular interest to researchers since the 1980s. While frequentist nonparametric and Bayesian parametric methods have been proposed to precisely estimate the entire distribution of reactive inhibition, quantified by stop signal reaction times (SSRT), there is no method yet in the stop signal task literature to precisely estimate the entire distribution of proactive inhibition. We identify the proactive inhibition as the difference of go reaction times for go trials following stop trials versus those following go trials and introduce an Asymmetric Laplace Gaussian (ALG) model to describe its distribution. The proposed method is based on two assumptions of independent trial type (go/stop) reaction times and Ex-Gaussian (ExG) models. Results indicated that the four parametric ALG model uniquely describes the proactive inhibition distribution and its key shape features, and its hazard function is monotonically increasing, as are its three parametric ExG components. In conclusion, the four parametric ALG model can be used for both response inhibition components and its parameters and descriptive and shape statistics can be used to classify both components in a spectrum of clinical conditions.

## 1. Introduction

### 1.1. Stop Signal Task and the Race Model

Response inhibition refers to one’s ability to stop responses or impulses that have become inappropriate or unwanted within continually changing environments [1]. This process’s importance lies in one’s being in continually changing conditions that require new, updated courses of action [2]. Some instances of response inhibition in daily life include braking while driving a vehicle into an intersection in reaction to a sudden traffic change, changing direction during a tennis game and resisting an extra piece of pizza at a birthday party. Two paradigms have been proposed to study the lab setting’s response inhibition [3]: the stop signal task and the go/no-go task. In the standard stop signal task, as used in this study, the task consists of a two-choice response time task called the “go task” and the “stop task”. The go task is the primary task in which the participants are asked to correctly press a right or left button in response to stimulus presentation—an “X” or “O” on the computer screen. The stop task is the occasional, secondary task in which (with a probability of stop signal $p_{ss}$) the participants are presented with a stop signal alarm after a temporal delay; participants are instructed to withhold their responses to the ongoing go task. Successful response inhibition occurs when participants successfully withhold their response to the “X” or “O” on the screen in the stop task (Figure 1).

Within the Stop Signal Task (SST) paradigm, response inhibition has been evaluated and described by several methods, such as the deadline model, the race model (with its independent and interactive versions) and the Hans–Carpenter model [5,6,7,8,9]. This work considers the independent race model given its widespread application in the SST literature. The independent race model creates a context to measure one of the inhibition components: the latency of unobservable stopping response or Stop Signal Reaction Times (SSRT). The model is based on two competing processes: the go process ($T_{go}$) and the stop process ($T_{stop}$). In all experiment trials, the go process starts upon triggering the stimulus, and, in a portion of them—called stop trials—after the stop signal delay times ($T_{d}$) following the stimulus, the stop process starts. The stop process wins (or loses) the race against the go process whenever $T_{go}>T_{stop}+T_{d}\left(T_{go}<T_{stop}+T_{d}\right),$ and in this case, successful inhibition (or failed inhibition) occurs.

The independent race model assumes two independent assumptions: first, an stochastic independence assumption between the go process and the stop process (i.e., $T_{go}⊥T_{stop}$); second, a contextual independence assumption, meaning that the go process has the same distribution over all SST trials (i.e., $\left(T_{go}|T_{d}\right)=T_{go}$). The validity of these assumptions has been the subject of various discussions in the SST literature [10].

The outline of the introduction in the subsequent sections is as follows. In Section 1.2, we introduce two major components of inhibition: the reactive inhibition and proactive inhibition. Next, we review different contextual index type (constant vs. distribution) measurements of reactive inhibition. Then, in Section 1.3, we focus on contextual estimations of proactive inhibition. Here, in Section 1.3.1, we review its main constant indices, and, in Section 1.3.2, we discuss the motivation for the distributional index. Finally, in Section 1.4, we outline the remainder of this work.

### 1.2. Components of Inhibition

Response inhibition has two distinctive temporal–dynamic components: reactive inhibition and proactive inhibition. Both components have been utilized within the standard stop signal task, or its varieties, to discriminate different clinical groups [11,12]. We generally refer to reactive inhibition as the outright inhibition triggered by an external cause. It is the reactive cancellation upon recognizing environmental changes that require stopping behavior. Next, proactive inhibition is the restraint of actions in preparation for stopping by external conditions [13]. It involves slowing responses and monitoring for the need to stop. Each of these inhibition components has been quantified in distinctive methods as constant point estimates or distributions in the stop signal task (SST) literature [1,14,15,16,17,18,19,20,21,22,23,24,25,26,27,28,29]. Figure 2 summarizes the inhibition components and their index types.

Reactive inhibition (Figure 2: path 1-1, 1-2-1, 1-2-2-1, 1-2-2-2) has been quantified as Stop Signal Reaction Times (SSRT) in the SST literature from both point estimation and distributional perspectives. Primary point estimations of the reactive inhibition to date include the Crude SSRT, the Logan 1994 SSRT [1], the Weighted SSRT, the Mixture SSRT [24] and the time series-based SSRT [26]. On the other hand, primary distributional estimations of the reactive inhibition include Colonius’s nonparametric method [29], the Bayesian Parametric Approach (BPA) [19,20]—with two subtype methods: the Individual BPA (IBPA) and the Hierarchical BPA (HBPA)—and the mixture method [25]. In the case of parametric mixture SSRT, the cluster type components may take a variety of proposed reaction time (RT) models, such as Ex-Gaussian (ExG), Ex-Wald, Wald, Gamma, Weibull and lognormal [21,23,27]. However, given the ExG model’s practical advantages to others, the parametric model is widely considered for the reactive inhibition [19,20].

In healthy subjects, proactive phases of the Stop Signal Task (SST) selectively activate right frontoparietal regions involved in proactive withholding, and reactive phases selectively activate right inferior frontal and caudate regions involved in the reactive cancellation of responses [30,31,32,33]. However, activities during proactive and error phases of the task strongly predict SSRT in controls and SSRT differences in ADHD [31,34]. Similarly, it is most likely that activities at times other than the proactive inhibition phase strongly contribute to estimates of proactive control. As such, observed differences in proactive control cannot be directly attributed to specific brain regions or phases of activity. The term “proactive inhibition” has several interpretations within the stop signal task literature: first, “the advanced preparation to halt action in the anticipation of an imminent stop signal” [16]; second, “association of probability of stopping in the stop trial and the go reaction times (GORT)” [35]; third, “comparison of two go trial reaction times (GORT1,GORT2) before and after a go trial as well as before and after a stop trial “ [36]; fourth, “comparison of go trial reaction times after a stop trial (GORTB) to that after a go trial(GORTA)” [25]. In this paper, we operationalize the last interpretation, which has the advantage of requiring only one single-arm SST.

### 1.3. Estimations of Proactive Inhibition

#### 1.3.1. Constant Index

Proactive inhibition has been quantified in the SST literature merely as point estimation from three different perspectives (Figure 2. path 2-1). The first such estimation is defined through a dynamic Bayesian model [18]. Here, using the mixture assumption for the predictive distribution for the probability of the kth trial being a stop task “p(Stop)”, the reactive inhibition is defined as the positive Pearson correlation called the “Sequential Effect (SE)” (see Equation (1)). The second estimation is based on the variation of differences of reaction times in a go trial (GORTs) in the associated arms of the modified standard stop signal task paradigm [14,15,16,17,22,28]. The associated arm of the study is identified by the set of all go and stop trials associated with the specific pre-designed probability of stop signal $p_{ss}$ (e.g., as in [14]: one may run four arms of 60, 200, 120 and 86 trials with $p_{ss}=0,0.15,0.25$ and 0.35, respectively). Generally, for two given probabilities of stop signals in two arms of the modified SST where $0\leq p_{ss}^{\left(1\right)}<p_{ss}^{\left(2\right)}<1,$ the arm type proactive inhibition $\Delta_{AT}GORT$ is defined as the difference of mean GORT in the two arms (see Equation (2)). The last type of the point estimation of proactive inhibition is based on the differences of GORTs in the trial type clusters of the standard SST paradigm [24,26] (Figure 3). Here, for the fixed stop signal probability (e.g., $p_{ss}=0.25$) and the type A GORTA and type B GORTB, the trial type proactive inhibition $\Delta_{TT}GORT$ is defined as the difference of mean GORT in these two types (see Equation (3)).

The following equations summarize the formulae for the three methods, respectively:(1)$$\begin{array}{ccc} SE & = & Corr(p(Stop),GORT), \end{array}$$(2)$$\begin{array}{ccc} \Delta_{AT}GORT & = & mean\left(GORT_{\left(p_{ss}^{\left(2\right)}\right)}\right)-mean\left(GORT_{\left(p_{ss}^{\left(1\right)}\right)}\right), \end{array}$$(3)$$\begin{array}{ccc} \Delta_{TT}GORT & = & mean\left(GORT_{B}\right)-mean\left(GORT_{A}\right). \end{array}$$

#### 1.3.2. Motivation

Little information is available in the SST literature on the entire distribution of proactive inhibition and its key features (Figure 2: path 2-2-1; 2-2-2). This subject is significant for two reasons: first, although what we know about brain function prevents a mapping of discrete behavioral observables onto discrete phases of modular brain functions, it is also conversely true that proactive inhibition distributions reflect information far beyond isolated brain activity during isolated phases of SST trials and therefore have strong potential to complement information available from brain imaging. Furthermore, it might reveal important clinical differences not available from brain imaging (which is already known to be the case for point estimates of SSRT). Second, measures of central tendencies, such as mean or median, are insufficient—and even unnecessary—to compare mostly skewed response inhibition distributions [37]. Besides this, masking prominent features of the proactive inhibitions by using central tendency measures may result in incorrect conclusions about their make-up. As an example, two different clinical groups may have the same mean of proactive inhibition, but the shape of their distributions may differ: one may be more positively skewed, or more leptokurtic, or possess a higher domain of variance. The methods mentioned earlier for the estimation of proactive inhibition do not allow for the precise estimation and description of the appropriate models for the entire set of proactive inhibition distributions.

### 1.4. Study Outline

This study proposes a four parametric, Asymmetric Laplace Gaussian (ALG) model for the entire proactive inhibition, given the assumption of independent trial type (go/stop) GORTs within the standard stop signal task (Figure 2, path 2-2-2-2). The study outline is as follows. First, in Section 2, some mathematical preliminaries on the components of ALG distribution, their features, the definition of the proactive inhibition distribution, and the involved random variables in the stop signal task probability space are provided. Second, in Section 3, a mathematical analysis regarding the distribution of proactive inhibition is presented. This includes its four parametric ALG form, descriptive statistics, shape statistics and vital distributional properties of the ALG model (e.g., shape and tail behavior, hazard function). Finally, an empirical example is presented to manifest the above theoretical results. Here, as in [24,26], the overall SST data for each participant are partitioned to type A SST data and type B SST data. Then, using the Individual Bayesian Parametric Approach (IBPA) method [19,20], the fitted ExG GORT mean posterior parameters are calculated for the cluster type SST data. Next, the distribution of proactive response inhibition is studied. Table 1 presents the summary of estimation methods presented in the literature, including this study, given the type of inhibition component.

## 2. Mathematical Preliminaries

### 2.1. Preliminaries on Component Distributions

The reader who has studied intermediate probability is well-equipped with the following definitions and theorems. A good comprehension of them plays a key role in understanding the nature of the proposed ALG model and its relationship dynamic with its ExG components. The first two definitions provide the critical features of the components of our upcoming calculations, namely Ex-Gaussian distribution (ExG) [38] and the Asymmetric Laplace distribution (AL) [39]. The third definition deals with the Asymmetric Laplace Gaussian (ALG) distribution. The subsequent four theorems play key roles in the proofs of the properties of the ALG model in Section 3.1.

**Definition** **1.**
*A random variable has an Ex-Gaussian (ExG) distribution with parameters (μ,σ,τ) whenever it is considered as the sum of an independent normal random variable with parameters $\left(\mu ,\sigma^{2}\right)$ and an exponential random variable with parameter τ:*

(4)
$$\begin{array}{ccc} ExG(\mu ,\sigma ,\tau ) & \overset{d}{=} & N\left(\mu ,\sigma^{2}\right)⊕Exp\left(\tau \right). \end{array}$$



The density, the moment generating function, the nth cumulant (n≥1), the variance, the skewness and the kurtosis of the ExG distribution are given by
(5)$$\begin{array}{ccc} PDF & & f_{ExG}\left(t|\mu ,\sigma ,\tau \right)=\frac{1}{\tau }exp\left(\frac{\mu -t}{\tau }+\frac{\sigma^{2}}{2\tau^{2}}\right)*\Phi \left(\frac{\mu -t}{\sigma }-\frac{\sigma }{\tau }\right):\sigma ,\tau >0,t\in R,\\ MGF & & m_{ExG}\left(t\right)={\left(1-t\tau \right)}^{-1}exp\left(\mu t+\frac{\sigma^{2}}{2}t^{2}\right):t<\tau^{-1},\\ n^{th}Cumulant & & \kappa_{n}^{ExG}=\left(n-1\right)!\tau^{n}+1_{n=1}\left(n\right)\mu +1_{n=2}\left(n\right)\sigma^{2}:1\leq n,\\ Mean & & E(ExG)=\mu +\tau ,\\ Variance & & Var\left(ExG\right)=\sigma^{2}+\tau^{2},\\ Skewness & & \gamma_{ExG}=2{\left(1+\sigma^{2}\tau^{-2}\right)}^{-3/2},\\ Kurtosis & & \kappa_{ExG}=3\frac{\left(1+2\sigma^{-2}\tau^{2}+3\sigma^{-4}\tau^{4}\right)}{{\left(1+\sigma^{-2}\tau^{2}\right)}^{2}}. \end{array}$$

**Definition** **2.**
*A random variable has an Asymmetric Laplace (AL) distribution with parameters $\left(\alpha_{1},\alpha_{2}\right)$ whenever it is considered as the difference of two independent exponential random variables with parameters $\alpha_{2}$, and $\alpha_{1}$, respectively:*

(6)
$$\begin{array}{ccc} AL(\alpha_{1},\alpha_{2}) & \overset{d}{=} & Exp\left(\alpha_{2}\right)⊖Exp\left(\alpha_{1}\right). \end{array}$$



The density, the moment generating function, the nth cumulant (n≥1), the variance, the skewness and the kurtosis of the AL distribution are given by
(7)$$\begin{array}{ccc} PDF & & f_{AL}\left(t|\alpha_{1},\alpha_{2}\right)=\frac{exp\left(\frac{t}{\alpha_{1}}\right)1_{\left(-\infty ,0\right)}\left(t\right)+exp\left(\frac{-t}{\alpha_{2}}\right)1_{\left[0,\infty \right)}\left(t\right)}{\alpha_{1}+\alpha_{2}}\;t\in R,\\ MGF & & m_{AL}\left(t\right)={\left(1+\left(\alpha_{1}-\alpha_{2}\right)t-\alpha_{1}\alpha_{2}t^{2}\right)}^{-1}\;-\alpha_{1}^{-1}<t<\alpha_{2}^{-1},\\ n^{th}Cumulant & & \kappa_{n}^{AL}=\left(n-1\right)!\left({\left(-\alpha_{1}\right)}^{n}+{\left(\alpha_{2}\right)}^{n}\right)\;1\leq n,\\ Mean & & E\left(AL\right)=-\left(\alpha_{1}-\alpha_{2}\right),\\ Variance & & Var\left(AL\right)=\alpha_{1}^{2}+\alpha_{2}^{2},\\ Skewness & & \gamma_{AL}=-2\left(\alpha_{1}^{3}-\alpha_{2}^{3}\right)\times {\left(\alpha_{1}^{2}+\alpha_{2}^{2}\right)}^{-3/2},\\ Kurtosis & & \kappa_{AL}=3\left(3\alpha_{1}^{4}+2\alpha_{1}^{2}\alpha_{2}^{2}+3\alpha_{2}^{4}\right)\times {\left(\alpha_{1}^{2}+\alpha_{2}^{2}\right)}^{-2}. \end{array}$$

The convolution of two independent AL random variables and Gaussian random variables, called ALG or Normal-Laplace (NL) random variables, has been of special attention in the literature [40,41]. We adopt ALG notation in this work given its alignment with the ExG notation.

**Definition** **3.**
*A random variable has Asymmetric Laplace-Gaussian (ALG) distribution with parameters $\left(\alpha_{1},\alpha_{2},\mu ,\sigma \right)$ whenever it is considered as the sum of two independent Asymmetric Laplace random variables with parameters $\left(\alpha_{1},\alpha_{2}\right)$ and a Normal random variable with parameters $\left(\mu ,\sigma^{2}\right)$, respectively:*

(8)
$$\begin{array}{ccc} ALG(\alpha_{1},\alpha_{2},\mu ,\sigma ) & \overset{d}{=} & AL\left(\alpha_{1},\alpha_{2}\right)⊕N\left(\mu ,\sigma^{2}\right). \end{array}$$



Note that since $AL\left(0^{+},\alpha_{2}\right)\overset{d}{=}Exp\left(\alpha_{2}\right),$ it follows that
$$ALG\left(0^{+},\alpha_{2},\mu ,\sigma \right)\overset{d}{=}ExG\left(\mu ,\sigma ,\alpha_{2}\right).$$

Consequently, the ExG model can be considered a special degenerate ALG model. Next, the following key two theorems allow us to propose the ALG distribution as the model for the proactive inhibition and compute the key descriptive and shape statistics of the ALG distribution in terms of its Laplacian and Gaussian components [42].

**Theorem** **1.**
*Let X,Y be two independent real-valued random variables with finite moment generating functions $m_{X},m_{Y},$ and cumulant functions κ,κ, respectively. Then, for some $s_{0}>0,$*

(9)
$$m_{X+Y}\left(t\right)=m_{X}\left(t\right)m_{Y}\left(t\right):\;\;\left(-s_{0}<t<s_{0}\right),$$


(10)
$$\kappa_{X+Y}\left(t\right)=\kappa_{X}\left(t\right)+\kappa_{Y}\left(t\right):\;\;\;\left(-s_{0}<t<s_{0}\right).$$



**Theorem** **2.**
*Let X,Y be two real-valued random variables with finite moment generating functions $m_{X},m_{Y},$ respectively. Assume for some $s_{0}>0:\;m_{X}\left(t\right)=m_{Y}\left(t\right)\;\left(-s_{0}<t<s_{0}\right).$ Then, X,Y have the same distribution.*


Finally, the last two theorems enable us to describe the behavior of the hazard function of the ALG model for the proactive inhibition [43,44].

**Theorem** **3.**
*Let X be a real-valued random variable with differentiable PDF $f_{X}$ and CDF $F_{X}$ such that $f_{X}\left(t\right)\to 0,F_{X}\left(t\right)\to 1\;as\;t\to \infty ,$ and $-ln(f_{X}\left(t\right))$ is convex (concave). Then, the hazard function $h_{X}$ is increasing (decreasing).*


**Theorem** **4.**
*Let X,Y be two independent, real-valued random variables with non-decreasing hazard functions $h_{X},h_{Y},$ respectively. Then, the hazard function of their sum $h_{X+Y}$ is non decreasing as well.*


### 2.2. Stop Signal Task Probability Space and Random Variables

Given the stop signal task described in Section 1.1, its associated probability space (Ω,F,P) constitutes of a sample space Ω as the set of all go trials, F as the power set of sample space (F=P(Ω)) and a probability measure *P* defined by $P\left(E\right)=\int_{E}dF,\;E\in F$ where *F* is the cumulative distribution function (CDF) of some RT distribution (e.g., ExG). Furthermore, the go trial RT random variable is a measurable function GORT:Ω→R. Moreover, for specific type-A (all trials immediately following go trials), type-B (all trials immediately following stop trials) and type-S (all trials) cluster SST data, their associated probability spaces and random variables are denoted by the associated subscripts (e.g., $\left(\Omega_{A},F_{A},P_{A}\right),\left(\Omega_{B},F_{B},P_{B}\right)$ and $\left(\Omega_{S},F_{S},P_{S}\right)$). In order to have dimensional homogeneity [45] on having algebraic operations over RT random variables $GORT_{A}:\Omega_{A}\to R$ and $GORT_{B}:\Omega_{B}\to R$, one may consider extension via projection to a higher dimensional sample space or coupling [46]. Considering the first method, both cluster type probability spaces $\left(\Omega_{A},F_{A},P_{A}\right)$ and $\left(\Omega_{B},F_{B},P_{B}\right)$ can be simultaneously extended to the following probability space:(11)$$\begin{array}{ccc} \left(\Omega_{A\times B},F_{A⊗B},P_{\pi }\right) & = & \left(\Omega_{A}\times \Omega_{B},P\left(\Omega_{A}\right)⊗P\left(\Omega_{B}\right),P_{A}\times P_{B}\right) \end{array}$$
where $\Omega_{A}\times \Omega_{B}$ is the Cartesian product of cluster type sample spaces; $P\left(\Omega_{A}\right)⊗P\left(\Omega_{B}\right)$ is the tensor product sigma algebra on product space; and $P_{A}\times P_{B}$ is the canonical product probability measure. In particular, considering the following projections:$$\begin{array}{ccc} \pi_{A}:\Omega_{A}\times \Omega_{B}\to \Omega_{A} & : & \pi_{A}\left(g_{A},g_{B}\right)=g_{A},\pi_{A}^{-1}\left(E_{A}\right)\in F_{A⊗B}\;for\;all\;E_{A}\in F_{A}\\ \pi_{B}:\Omega_{A}\times \Omega_{B}\to \Omega_{B} & : & \pi_{B}\left(g_{A},g_{B}\right)=g_{B},\pi_{B}^{-1}\left(E_{B}\right)\in F_{A⊗B}\;for\;all\;E_{B}\in F_{B} \end{array}$$
it follows that:$$\begin{array}{ccc} P_{\pi }\left(\pi_{A}^{-1}\left(E_{A}\right)\right) & = & P_{A}\left(E_{A}\right)\;for\;all\;E_{A}\in F_{A}\\ P_{\pi }\left(\pi_{B}^{-1}\left(E_{B}\right)\right) & = & P_{B}\left(E_{B}\right)\;for\;all\;E_{B}\in F_{B}. \end{array}$$

The cluster type space-related random variables $GORT_{A}:\Omega_{A}\to R$ and $GORT_{B}:\Omega_{B}\to R$ can now be extended to the corresponding random variables $GORT_{A\times B}^{\pi }:\Omega_{A\times B}\to R\;\left(GORT_{A}^{\pi }\left(g_{A},g_{B}\right)=GORT_{A}\left(g_{A}\right)\right)$ and $GORT_{A\times B}^{\pi }:\Omega_{A\times B}\to R\;\left(GORT_{B}^{\pi }\left(g_{A},g_{B}\right)=GORT_{B}\left(g_{B}\right)\right),$ respectively. In the spirit of this extension, the algebraic operations (e.g., difference) on the random variables $GORT_{A},GORT_{B}$ from different cluster type probability spaces will be defined as their corresponding higher dimensional extensions (e.g., $GORT_{B}\left(g_{B}\right)-GORT_{A}\left(g_{A}\right)=GORT_{B}^{\pi }\left(g_{A},g_{B}\right)-GORT_{A}^{\pi }\left(g_{A},g_{B}\right):\;for\;all\;\left(g_{A},g_{B}\right)\in \Omega_{A}\times \Omega_{B}$).

### 2.3. Proactive Inhibition Index

Proactive inhibition was operationalized based on the standard stop signal task’s internal perspective [26]. The following definition of distribution of proactive inhibition is inspired by the third constant index of proactive inhibition in Equation (3). Here, for a given fixed stop signal probability (e.g., 0.25), type A GORT of $GORT_{A}$ (GORT for a trial after a go trial) and type B GORT of $GORT_{B}$ (GORT for a trial after a stop trial), the internal proactive inhibition is defined as
(12)$$\begin{array}{ccc} \Delta GORT & \overset{d}{=} & GORT_{B}-GORT_{A}. \end{array}$$

**Remark** **1.**
*Equation (12) in the definition of distribution proactive inhibition has a direct relationship with Equation (3) in the definition of the constant index of proactive inhibition. Indeed, taking expectations from both sides of Equation (12) yields Equation (3), while considering constants as point mass distributions implies that Equation (3) is a special case of Equation (12).*


Note that there are two mathematical perspectives for the proactive inhibition: first, a model with two ExG components; second, a model with Asymmetric Laplace (AL) and Gaussian components. Henceforward, it is understood within the given context which perspective is being discussed.

## 3. Results

The results are discussed in two subsections. In Section 3.1, we explore the mathematical analysis of the proposed model for the proactive inhibition in the standard stop signal task. The proofs of the key results are presented in Appendix A. This model includes a four parametric ALG for the proactive inhibition and its prominent distributional properties. In Section 3.2, we first present an empirical example of the case and discuss its various distributional features. Then, we compare the proactive inhibition ALG model and the reactive inhibition ExG model in terms of key statistics.

### 3.1. Mathematical Analysis

#### 3.1.1. The Proactive Inhibition Distribution and its Parameters

First of all, we propose a mathematical model for the proactive inhibition provided by the ALG:

**Theorem** **5**(The Main Result). *The four parametric $ALG(\tau_{A},\tau_{B},\mu_{B}-\mu_{A},{\left(\sigma_{B}^{2}+\sigma_{A}^{2}\right)}^{1/2})$ may present a unique statistical model for the internal proactive inhibition index ΔGORT with trial type-related parameters $\left(\mu_{A},\sigma_{A},\tau_{A},\mu_{B},\sigma_{B},\tau_{B}\right)$ in the standard stop signal task.*

As a corollary of Theorem 5, the probability density function $\left(f_{\Delta GORT}\right)$ and the cumulative density function $\left(F_{\Delta GORT}\right)$ of the ALG distribution for the internal proactive inhibition index are given by [40,41]:(13)$$\begin{array}{ccc} f_{\Delta GORT}\left(t\right) & = & \frac{1}{\tau_{A}+\tau_{B}}\\ & \times [ & e^{\left(\frac{\sqrt{\sigma_{B}^{2}+\sigma_{A}^{2}}}{2\tau_{B}}\left(\frac{\sqrt{\sigma_{B}^{2}+\sigma_{A}^{2}}}{\tau_{B}}-2\frac{t-(\mu_{B}-\mu_{A})}{\sqrt{\sigma_{B}^{2}+\sigma_{A}^{2}}}\right)\right)}\times \left(1-\Phi \left(\frac{\sqrt{\sigma_{B}^{2}+\sigma_{A}^{2}}}{\tau_{B}}-\frac{t-(\mu_{B}-\mu_{A})}{\sqrt{\sigma_{B}^{2}+\sigma_{A}^{2}}}\right)\right)\\ & + & e^{\left(\frac{\sqrt{\sigma_{B}^{2}+\sigma_{A}^{2}}}{2\tau_{A}}\left(\frac{\sqrt{\sigma_{B}^{2}+\sigma_{A}^{2}}}{\tau_{A}}+2\frac{t-(\mu_{B}-\mu_{A})}{\sqrt{\sigma_{B}^{2}+\sigma_{A}^{2}}}\right)\right)}\times \left(1-\Phi \left(\frac{\sqrt{\sigma_{B}^{2}+\sigma_{A}^{2}}}{\tau_{A}}+\frac{t-(\mu_{B}-\mu_{A})}{\sqrt{\sigma_{B}^{2}+\sigma_{A}^{2}}}\right)\right)]\\ & & \;t\in R \end{array}$$
and,
(14)$$\begin{array}{ccc} F_{\Delta GORT}\left(t\right) & = & \frac{1}{\tau_{A}^{-1}+\tau_{B}^{-1}}\times [\left(\tau_{A}^{-1}+\tau_{B}^{-1}\right)\Phi \left(\frac{t-(\mu_{B}-\mu_{A})}{\sqrt{\sigma_{B}^{2}+\sigma_{A}^{2}}}\right)\\ & - & \tau_{A}^{-1}e^{\left(\frac{\sqrt{\sigma_{B}^{2}+\sigma_{A}^{2}}}{2\tau_{B}}\left(\frac{\sqrt{\sigma_{B}^{2}+\sigma_{A}^{2}}}{\tau_{B}}-2\frac{t-(\mu_{B}-\mu_{A})}{\sqrt{\sigma_{B}^{2}+\sigma_{A}^{2}}}\right)\right)}\times \left(1-\Phi \left(\frac{\sqrt{\sigma_{B}^{2}+\sigma_{A}^{2}}}{\tau_{B}}-\frac{t-(\mu_{B}-\mu_{A})}{\sqrt{\sigma_{B}^{2}+\sigma_{A}^{2}}}\right)\right)\\ & + & \tau_{B}^{-1}e^{\left(\frac{\sqrt{\sigma_{B}^{2}+\sigma_{A}^{2}}}{2\tau_{A}}\left(\frac{\sqrt{\sigma_{B}^{2}+\sigma_{A}^{2}}}{\tau_{A}}+2\frac{t-(\mu_{B}-\mu_{A})}{\sqrt{\sigma_{B}^{2}+\sigma_{A}^{2}}}\right)\right)}\times \left(1-\Phi \left(\frac{\sqrt{\sigma_{B}^{2}+\sigma_{A}^{2}}}{\tau_{A}}+\frac{t-(\mu_{B}-\mu_{A})}{\sqrt{\sigma_{B}^{2}+\sigma_{A}^{2}}}\right)\right)]\\ & & \;t\in R, \end{array}$$
respectively. Here, Φ denotes the standard normal cumulative distribution function.

Next, given trial type parameters, we estimate the descriptive and shape statistics for the proposed ALG model of the proactive inhibition.

**Theorem** **6.**
*The descriptive statistics and the shape statistics of the proactive inhibition ALG distribution with trial type-related parameters $\left(\mu_{A},\sigma_{A},\tau_{A},\mu_{B},\sigma_{B},\tau_{B}\right)$ in the standard stop signal task are given by*

(15)
$$\begin{array}{ccc} n^{th}Cumulant & & \kappa_{n}^{ALG}=\left(n-1\right)!\left({\left(-\tau_{A}\right)}^{n}+\tau_{B}^{n}\right)\\ & & \;+1_{\left(n=1\right)}\left(n\right)\left(\mu_{B}-\mu_{A}\right)+1_{\left(n=2\right)}\left(n\right)\left(\sigma_{B}^{2}+\sigma_{A}^{2}\right):\;\;1\leq n\\ Mean & & E\left(ALG\right)=\tau_{B}-\tau_{A}+\mu_{B}-\mu_{A},\\ Variance & & Var\left(ALG\right)=\tau_{A}^{2}+\tau_{B}^{2}+\sigma_{A}^{2}+\sigma_{B}^{2},\\ Skewness & & \gamma_{ALG}=\frac{2(\tau_{B}^{3}-\tau_{A}^{3})}{{\left(\tau_{A}^{2}+\tau_{B}^{2}+\sigma_{A}^{2}+\sigma_{B}^{2}\right)}^{3/2}},\\ Kurtosis & & \kappa_{ALG}=\frac{6(\tau_{B}^{4}+\tau_{A}^{4})}{{\left(\tau_{A}^{2}+\tau_{B}^{2}+\sigma_{A}^{2}+\sigma_{B}^{2}\right)}^{2}}. \end{array}$$



#### 3.1.2. The Proactive Inhibition’s Key ALG Distributional Properties

In this section, we present key distributional properties for the ALG model for proactive inhibition, including (i) component decompositions in terms of trial type GORT, (ii) shape and tail behavior and (iii) the behavior of the hazard function.

**Theorem** **7**(Component Decomposition). *An ALG model for proactive inhibition emerges from uncountable pairs of trial type related GORT $\left(GORT_{A},GORT_{B}\right)$ distributions.*

We remind the reader that Theorem 7 presents a process to simulate a plausible four parametric ALG distribution for the proactive inhibition.

**Theorem** **8**(Shape and Tail Behavior). *An ALG model for proactive inhibition has a unimodal, generally asymmetric, infinite, differentiable density with extreme large values proportionate to the $Exp(1/\tau_{B})$ distribution.*

We remind the reader that in contrast to the ALG model’s mean for proactive inhibition, there are no closed-form formulas for the mode and the median, respectively. Similar to the ExG model for reactive inhibition with increasing hazard function, we consider the following theorem.

**Theorem** **9**(Hazard Function’s Behavior). *An ALG model for proactive inhibition has a non-decreasing hazard function.*

### 3.2. The Empirical Example

This section presents an example of the empirical data and model for the theoretical results inferred in the previous section on the ALG model for proactive inhibition and its descriptive, shape and hazard function’s key features. These results are based on the cluster type IBPA estimation of mean posterior ExG parameters θ=(μ,σ,τ) presented in Table A1 (Section A.2).

**Remark** **2.**
*The reader is reminded that in the point estimation of the parameters of ALG, there is uncertainty inherited from random sampling in the Bayesian procedure [47]. The reader is recommended to assess this when applying the procedure and to be aware of the differences (see Section A.3).*


The proactive inhibition distribution estimation process follows these steps:Partition the Single SST data to type-A SST data and type-B SST data,Fit IBPA with underlying ExG assumption to type-A SST data and type-B SST data,Retrieve mean posterior ExG parameters to type-A GORT data and type-B GORT data,Plug the estimations in previous step into the ALG model presented in Theorem 5.

#### 3.2.1. Materials & Methods

##### Data

The study data have been previously described [24,25,26,48]. Data were collected at the Ontario Science Center in Toronto, Canada, in 2009–2010. Included were 16,099 participants aged 6 to 17. The participants’ parents provided the required ethical consent for the SST experiment. Each participant completed the SST task, including four blocks of 24 trials with a total of 96 trials, including random 25% stop trials (24 stops) and 75% go trials (72 goes). The SST tracking algorithm was designed so that, at the end of the trials, each participant achieved a 50% probability of successful inhibition.

##### Participants

The study participants are the same as those described in [25,26]. Included here is a unique subsample of 44 participants with a mean age of 12.1 years, with 96 SST trials for each, and an almost balanced number of trial type stop trials (10–14 type B stop trials vs. 1410 type A stop trials, respectively). This almost balanced number of trial type stop trials yields 14–10 type B go trials vs. 58–62 type A go trials, respectively.

##### The SST Clusters

The study stop signal task clusters have been described before in [24,25,26]. Each participant’s SST data were partitioned to type A SST data, where a go trial preceded all trials, and type B SST data, with all trials preceded by a stop trial. All four starting trials in their blocks were identified as type-A trials. Hence, each participant had three types of SST data clusters: Type-A SST cluster (i.e., 72 trials), Type B SST cluster (i.e., 24 trials), and Type-S SST cluster (all 96 trials). Then, using IBPA, the parameters of the corresponding Ex-Gaussian (ExG) GORT’s parameters (i.e., $\theta_{A}=\left(\mu_{A},\sigma_{A},\tau_{A}\right),\theta_{B}=\left(\mu_{B},\sigma_{B},\tau_{B}\right),\theta_{S}=\left(\mu_{S},\sigma_{S},\tau_{S}\right)$ were computed as described in Section 3.2.2.

#### 3.2.2. Statistical Analysis

The statistical ALG model to describe the proactive inhibition distribution was presented using moment generating functions [40]. Next, the ALG model’s descriptive and shape statistics were computed in terms of parameters of the cluster type ExG components [42]. Finally, its hazard function behavior was theoretically inferred using its components’ parameters, [44].

The ExG components of the presented statistical model were estimated using the IBPA method [19,20]. As in [25], each participant had three IBPA associated ExG parametric estimations $\theta_{A}=\left(\mu_{A},\sigma_{A},\tau_{A}\right),\theta_{B}=\left(\mu_{B},\sigma_{B},\tau_{B}\right),$ and $\theta_{S}=\left(\mu_{S},\sigma_{S},\tau_{S}\right),$ associated to type-A cluster SST data, type-B cluster SST data and all SST data, respectively. These parameters were estimated as the posterior means of the associated following IBPA procedure with three chains, 5000 burn-ins within 20,000 simulations in Bayesian Ex-Gaussian Estimation of stop-signal RT distributions (BEESTS) 2.0 software [20]:
Data Individual Priors$GORT∼ExG(\mu_{go},\sigma_{go},\tau_{go})$
$SRRT∼ExG\left(\mu_{go},\sigma_{go},\tau_{go},\mu_{stop},\sigma_{stop},\tau_{stop},SSD\right)I_{\left[1,1000\right]}^{+}$$\;\mu_{go},\sigma_{go},\tau_{go}∼U\left[10,2000\right]$$SSRT∼ExG\left(\mu_{go},\sigma_{go},\tau_{go},\mu_{stop},\sigma_{stop},\tau_{stop},SSD\right)I_{\left[1,1000\right]}^{+}$$\;\mu_{go},\sigma_{go},\tau_{go}∼U\left[10,2000\right]$

Two sets of comparisons were conducted using paired *t*-tests (DescTools, *R* software version 4.0.0 [49]): first, a primary comparison between the cluster-type fitted parameter of the ExG distribution, the descriptive statistics and the shape statistics; second, secondary comparisons between the ALG model descriptive and shape statistics and its associated cluster-type ExG components.

#### 3.2.3. The ALG Model for Proactive Inhibition

The ExG model for each of the two components of proactive inhibition has the following key features presented by Table 2. First, while type-B μ, and τ parameters are significantly larger than their type-A counterparts, there is no difference for the σ parameter. In addition, the sample average proactive inhibition is 92.1 ms (95% CI = (69.4,114.9)). Second, both ExG components are positively skewed and leptokurtic. Finally, there is no significant difference between their trial-type skewness and their trial-type kurtosis, respectively. Figure 4a presents the trial type ExG modeled components of the ALG model.

The ALG model for proactive inhibition has the following features, given cluster type parameter estimations. First, as a primary corollary of Theorem 6, the model is positively skewed whenever $\tau_{B}>\tau_{A}.$ The negatively skewed and symmetric cases hold whenever the strict greater inequality > is replaced with < and =, respectively. According to data in Table A1 in the appendix, all three cases exist (case 10: positive skew; case 16: symmetric; case 11: negative skew).

Figure 4b presents all the mentioned three cases. Overall, the model is positively skewed given the results in Table 2. Second, as the second corollary of Theorem 6, the model is leptokurtic whenever ${\left(2\left(\tau_{A}^{4}+\tau_{B}^{4}\right)\right)}^{1/2}-\left(\tau_{A}^{2}+\tau_{B}^{2}\right)>\sigma_{A}^{2}+\sigma_{B}^{2}.$ The platykurtic and mesokurtic cases hold whenever the strict greater inequality > is replaced with < and =, respectively. In particular, for the case $\tau_{A}\approx \tau_{B},$ the model is always platykurtic. Overall, the model is platykurtic given results in Table 2. Third, while the ALG model’s standard deviation is larger than its two ExG’s components, its skewness and kurtosis are significantly smaller. Finally, the ALG model has a strictly increasing hazard function for various skewness cases, as mentioned in Theorem 9 and presented in Figure 4c.

#### 3.2.4. Proactive Inhibition ALG Model versus Reactive Inhibition ExG Model

This section compares the ALG distribution of the proactive inhibition and the ExG distribution of the reactive inhibition in descriptive and shapes statistics. Table 3 presents the corresponding statistics for both models. As it is seen, the proactive ALG inhibition distribution has a significantly lower mean, lower skewness, lower kurtosis and higher standard deviation than reactive inhibition ExG distribution. Futhermore, while the proactive inhibition ALG distribution is platykurtic, the corresponding reactive ExG distribution is leptokurtic. However, the two distributions are both positively skewed. Overall, the two distributions are significantly distinctive.

## 4. Discussion

### 4.1. Present Work

This paper presents a four parametric model for the entire proactive inhibition distribution—the ALG model. This model is based on the two independent ExG components fitted to the trial type GORTs. Considering ExG models as a degenerate ALG model, this work indicates that a four parametric ALG model can model both response inhibition components (see Figure 5).

The proposed ALG model for proactive inhibition has several important aspects. First, the model is based on the independent assumptions of GORTA and GORTB. Such speculation is an additional postulate to the structure of the SST data, and its validity needs extra investigation. Second, contrary to the point estimations of proactive inhibition in the form of the mean [26] or correlation [18], it presents the entire distribution of the proactive inhibition. Third, the ALG model for proactive inhibition is entirely distinctive from the ExG model for reactive inhibition in terms of the mean, standard deviation and kurtosis. This result provides more evidence on the distinction between proactive inhibition and reactive inhibition as a whole distribution. Fourth, similar to the ExG model for reactive inhibition, the ALG model for proactive inhibition is skewed to the right and has a monotonically increasing hazard function. Finally, the closed form ALG model for proactive inhibition with its associated ExG modeled components is the most plausible option at the moment. Other non-ExG RT models for its components (e.g., Gamma, Weibull, Lognormal, Wald and Ex-Wald) and, hence, their trial type differences do not yield to any known closed-form distribution for proactive inhibition. This issue makes the study of their probabilistic features difficult. This limitation is easily verifiable by repeating the proof of Theorem 5 based on the uniqueness of moment-generating functions for other non-ExG RT models for the trial type GORTs.

The proposed ALG model for the proactive inhibition is estimated in two different methods. First, one may use the Bayesian (or frequentist-based) methods to fit the ExG parametric models to their trial-type components and then estimate the four parametric ALG model using Theorem 5, as done in Section 3.2. Second, one may subtract the trial type GORTs and fit the ALG model directly to the differenced GORT data using Maximum Likelihood (ML) or Expectation-Maximization (EM) algorithms [50]. Simulations of the model follows the similar logic.

The proposed ALG model uniquely distinguishes the reactive inhibition and proactive inhibition distribution in terms of vital distributional features. Table 4 presents an overall comparison for proactive inhibition and reactive inhibition in terms of the ALG model (considering ExG as its particular case):

There are some limitations in the proposed ALG model for proactive inhibition. First, since GORTA data and GORTB data are unmatched, there is no way to calculate their correlation quantitatively. Hence, from a quantitative perspective, checking the validity of the assumption of independent GORTA and GORTB is difficult. Second, by its definition, proactive inhibition takes only non-negative values, while the presented ALG model takes negative values. Third, similar to the ExG model for reactive inhibition, the ALG model for proactive inhibition has a monotonically increasing hazard function, preventing it from being the best fitting model for the cases of proactive inhibition with peaked hazards. Finally, given the calculation structure of the ALG model parameters, based on those of ExG components, its parameters’ cognitive interpretations are highly dependent on its ExG components inheriting their constraints.

### 4.2. Future Work

Future research should replicate the proposed approach in modeling the proactive inhibition distribution in this study in different directions. This further work may include the following perspectives: first, the assumption of independence between $GORT_{A}$ and $GORT_{B}$ needs extra investigation and, in case of its violation, an updated model for proactive inhibition distribution is plausible. Second, one model should consider peaked hazard functions for the ALG model components to address RT data with such features. Third, there is a need to interpret the proposed ALG distribution parameters in terms of inhibition mechanisms in the brain and vice versa. Fourth, there is a lack of investigation comparing the proactive inhibition distribution in terms of the usual stochastic order, the descriptive and shape statistics across a spectrum of clinical groups such as ADHD, OCD, schizophrenia and drug users. Finally, similar investigations on comparing the proactive inhibition distribution and its above key statistics are plausible in terms of the participants’ ages.

### 4.3. Conclusions

In conclusion, the ALG model provides a practical description of the proactive inhibition distribution that takes full advantage of its ExG components fitted for the trial type GORTs. It also offers a straightforward, computational analog of the proactive inhibition, comparable to the ExG model for reactive inhibition. Given the advantages of estimating the entire distribution of proactive inhibition over former point estimations, the researchers recommend considering the ALG model as the latest optimal choice to describe the distribution of proactive inhibition.

## Figures and Tables

**Figure 1 brainsci-12-00730-f001:**
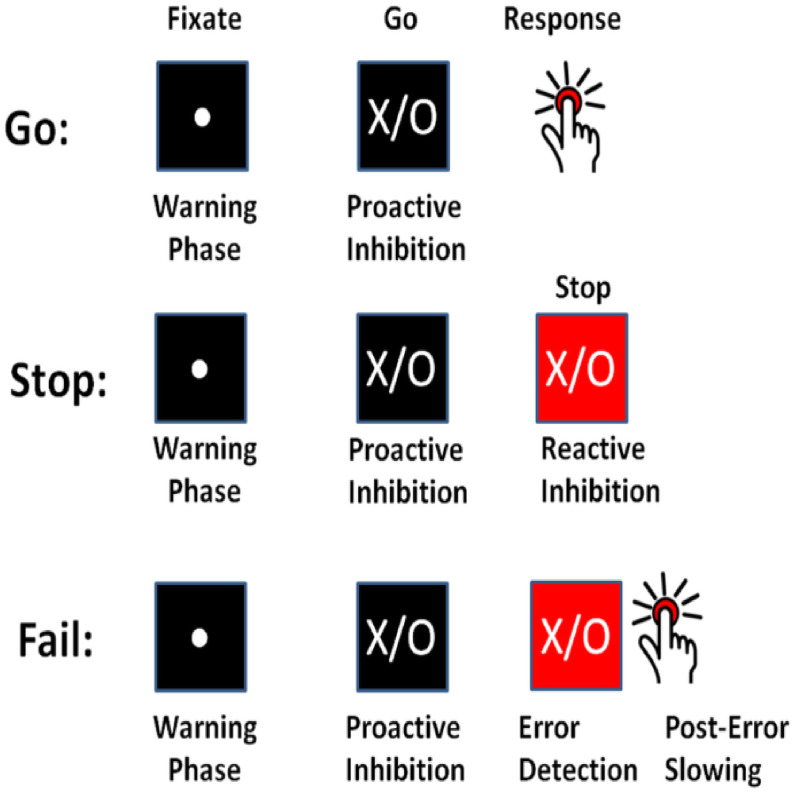
The standard stop signal task with two inhibition components: proactive inhibition, reactive inhibition (Chevrier and Schachar, 2020 [4]-Permission was granted).

**Figure 2 brainsci-12-00730-f002:**
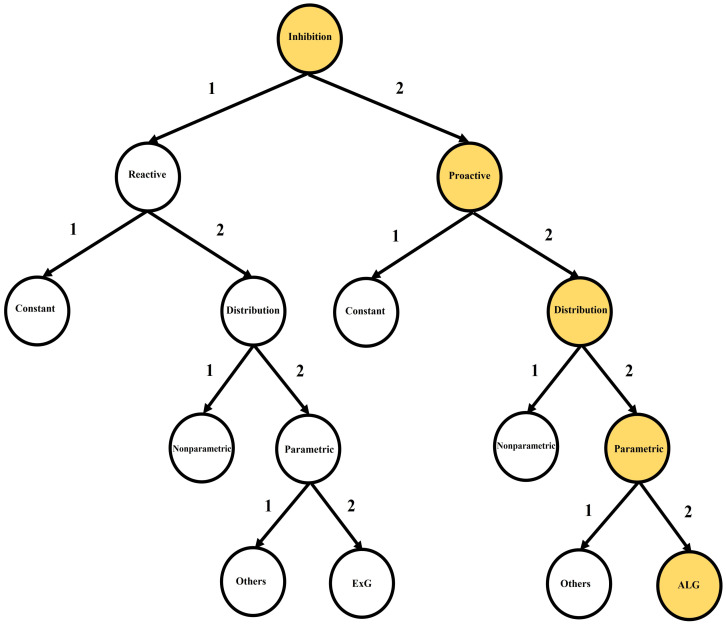
Inhibition components and their subtypes: current literature (path 1-1, 1-2-1, 1-2-2-1, 1-2-2-2, 2-1, 2-2-1, 2-2-2-1); this study (path 2-2-2-2).

**Figure 3 brainsci-12-00730-f003:**
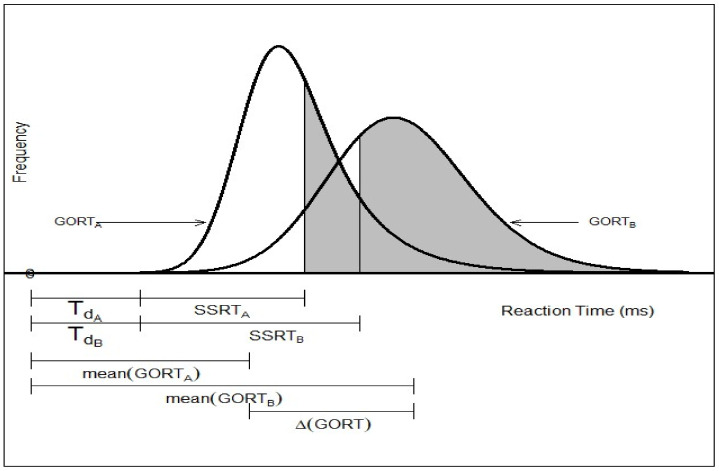
Trial type point estimation of proactive inhibition (Δ(GORT)) in the standard stop-signal task.

**Figure 4 brainsci-12-00730-f004:**
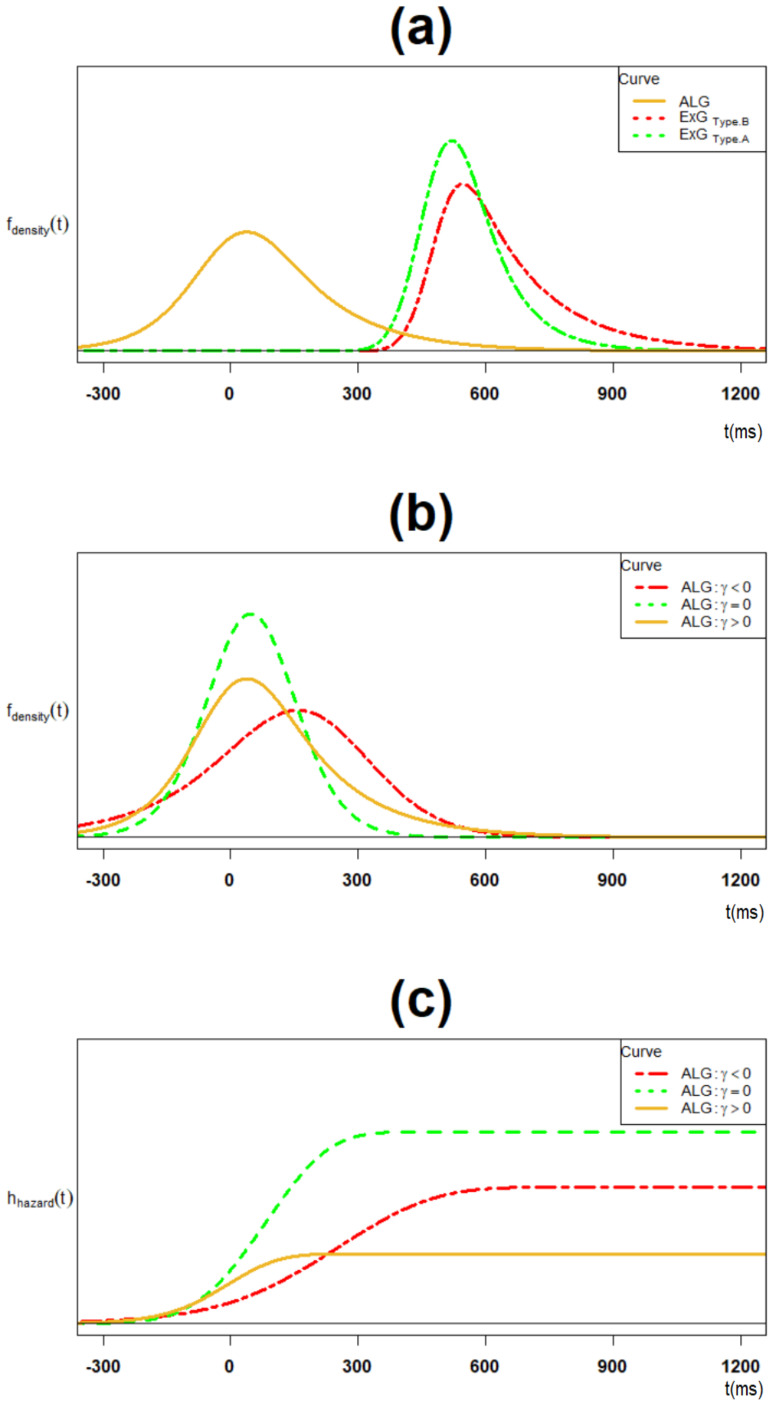
(**a**) The ALG density and its trial type ExG component densities; (**b**) the ALG density for the positively skewed, symmetric and negatively skewed cases; (**c**) the ALG hazard function for the positively skewed, symmetric and negatively skewed cases.

**Figure 5 brainsci-12-00730-f005:**
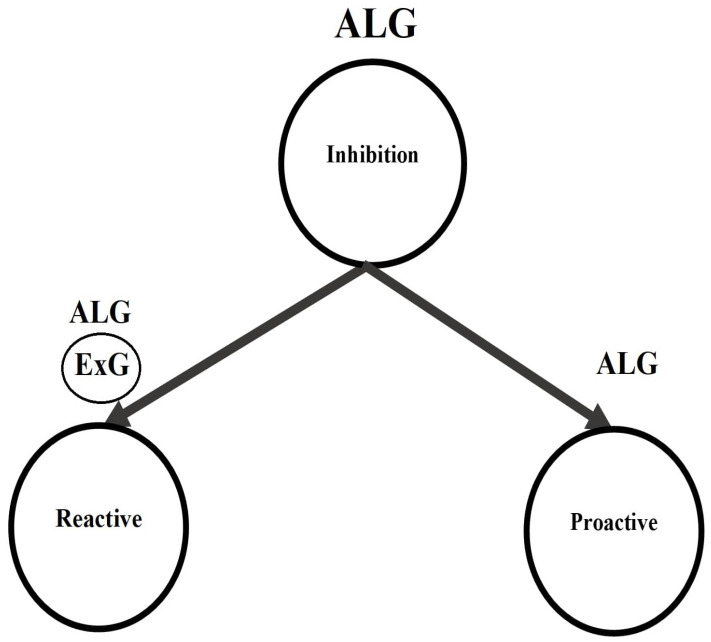
The ALG model as the comprehensive statistical model for inhibition in the standard SST.

**Table 1 brainsci-12-00730-t001:** Summary of Estimation Methods of Inhibition Components.

Estimation	Inhibition Component	
	**Reactive**	**Proactive**
Constant Index	SSRT	ΔGORT,SE
Examples	$SSRT_{Crude},SSRT_{Mixture},SSRT_{Weighted}$	$\Delta_{AT}GORT,\Delta_{TT}GORT$
	$SSRT_{Logan1994},SSRT_{SS.Logan1994}$	
Distribution Index	SSRT	ΔGORT
Examples	ExG,LN,Wald	ALG
	Ex-Wald, Gamma	

**Table 2 brainsci-12-00730-t002:** Descriptive and paired *t*-test [mean (95%CI)] results for parameters, descriptive and shape statistics of fitted Ex-Gaussian distribution to cluster type GORT and AL-Gaussian distribution to ΔGORT (n=44).

			ExG Model		ALG Model
		**Cluster**		**Comparison**	**Cluster**
		**Type A**	**Type B**	**Type B vs. Type A**	**Type S**
	$\alpha_{1}$	-	-	-	104.2
		-	-	-	(90.4, 117.9)
	$\alpha_{2}$	-	-	-	142.4
		-	-	-	(125.9, 158.8)
Parameter	μ	478.8	532.8	53.9 ***	53.9
		(448.0, 509.7)	(498.6, 566.9)	(30.9, 76.9)	(30.9, 76.9)
	σ	109.9	133.1	23.2	179.2
		(90.5, 129.3)	(108.4, 157.8)	(−0.1, 46.4)	(151.4, 206.9)
	τ	104.2	142.4	38.2 ***	-
		(90.4, 117.9)	(125.9, 158.8)	(19.6, 56.8)	-
	Mean	583.0	675.1	92.1 ***	92.1
		(553.0, 612.9)	(633.8, 716.4)	(69.4, 114.9)	(69.4, 114.9)
Statistics	St.D	160.6	202.4	41.8 ***	260.4
		(143.5, 177.8)	(177.9, 226.9)	(25.9, 57.6)	(232.3, 288.6)
	Skewness	0.787	0.918	0.131	0.186
		(0.602, 0.973)	(0.751, 1.085)	(−0.113, 0.375)	(0.076, 0.296)
	Kurtosis	4.966	5.300	0.334	1.153
		(4.414, 5.518)	(4.790, 5.808)	(−0.397, 1.064)	(0.923, 1.384)

Notes: *** *p*-value < 0.0005.

**Table 3 brainsci-12-00730-t003:** Comparison of proactive inhibition ALG model versus reactive inhibition ExG model in terms of descriptive and shape statistics (n=44).

Inhibition		Reactive	Proactive	Proactive vs. Reactive
Index		*SSRT*	ΔGORT	ΔGvs.S
Model		ExG	ALG	ALG vs. ExG
Statistics	Mean	196.8	92.1	−104.6 ***
		(173.5, 220.1)	(69.4, 114.9)	(−140.6, −68.7)
	St.D	157.8	260.4	102.6 ***
		(139.4, 176.2)	(232.3, 288.6)	(71.8, 133.6)
	Skewness	0.578	0.186	−0.401 ***
		(0.500, 0.674)	(0.076, 0.296)	(−0.540, −0.261)
	Kurtosis	4.231	1.153	−3.077 ***
		(3.998, 4.465)	(0.923, 1.384)	(−3.381, −2.775)

Notes: *** *p*-value < 0.0005.

**Table 4 brainsci-12-00730-t004:** Comparison of proactive inhibition and reactive inhibition in terms of ALG model properties.

Inhibition	Index	# Parameters	# Estimations	Mean	StD	Skewness (+)	Kurtosis	Hazard
Proactive	ΔGORT	4	2	lower	higher	lower	platykurtic	increasing
Reactive	SSRT	3	1	higher	lower	higher	leptokurtic	increasing

## Data Availability

Not applicable.

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
