# Peer review of "The Asymmetric Laplace Gaussian (ALG) Distribution as the Descriptive Model for the Internal Proactive Inhibition in the Standard Stop Signal Task"

_brainsci, 2022, doi:10.3390/brainsci12060730_

Round 1

Reviewer 1 Report

A new distribution is introduced to estimate inhibition distribution in SSRT framework. Theoretical results are supplemented with numerical results. Generally the article gives a good impression, mathematical results are sound (although mot all entirely new, as the authors  have already worked on the subject earlier), so that the article can be recommended for minor. There are a few points that might be taken care of.

  1. Results like Theorems 2.4-2.6 are standard and can be omitted, and replaced with standard references which are aplenty.
  2. A small language editing is needed to correct minor mistakes. Any reference or note within parenthesis should be with a space, e.g. convex (concave) . . . increasing (decreasing) etc. as in Theorem 2.6. L 187, P 7: following two key theorems; etc.
  3. Regarding the introduced distribution: First, why should an entire distribution be estimated, except if one is not typically interested in Bayesian analysis, i.e., why should a frequentist alternative not be considered? Second, since the main idea is introduced under independence assumptions, is it really so that no parametric methodology works for the comparison of two groups and a new distribution is needed (the authors say in abstract that no such method exists). At least this aspect has not been demonstrated, e.g. by comparison, by the authors. There is indeed a long list of distributions used to study cognitive behavior, particularly for group comparison. So the reader would be interested to know why a new model at all, and how it stands in relation of others. Moreover, the authors do compute moments and study other properties of the ALG distribution, but a rigorous proof of uniqueness claim made in abstract is missing.

Reviewer 2 Report

It is a well-written manuscript which can be of interest to the readers. The novel contributions need to be more clearly explained. Some of the theorems, especially theorems in section 2 are well-known in probability literature, and not novel derivations. These should be denoted as known results, rather than theorems in the paper, so that the results that the novel contributions are clear.
